# An international comparison of factors affecting quality of life among patients with congestive heart failure: A cross-sectional study

Brita Roy [1,2]*, Judith R. L. M. Wolf[3], Michelle D. Carlson[4,5], Reinier Akkermans[6,7], Bradley Bart[4,5], Paul Batalden[8], Julie K. Johnson[9], Hub Wollersheim[7], Gijs Hesselink[6]

1 Department of Internal Medicine, Section of General Internal Medicine, Yale School of Medicine, New Haven, CT, United States of America, 2 Department of Chronic Disease Epidemiology, Yale School of Public Health, New Haven, CT, United States of America, 3 Impuls - Netherlands Center for Social Care Research, Radboud Institute for Health Sciences, Radboud University Medical Center, Nijmegen, The Netherlands, 4 Minneapolis Veterans Administration Health Care System, Minneapolis, MN, United States of America, 5 University of Minnesota Medical School, Minneapolis, MN, United States of America, 6 IQ Health Care, Radboud Institute for Health Sciences, Radboud University Medical Center, Nijmegen, The Netherlands, 7 Department of Primary and Community Care, Radboud University Medical Center, Nijmegen, The Netherlands, 8 The Dartmouth Institute of Health Policy and Clinical Practice, Geisel Medical School at Dartmouth, Hanover, NH, United States of America, 9 Department of Surgery, Center for Healthcare Studies, Feinberg School of Medicine, Northwestern University, Chicago, IL, United States of America

* brita.roy@yale.edu

## Abstract

### Objective

To explore associations among twenty formal and informal, societal and individual-level factors and quality of life (QOL) among people living with congestive heart failure (CHF) in two settings with different healthcare and social care systems and sociocultural contexts.

### Setting and participants

We recruited 367 adult patients with CHF from a single heart failure clinic within two countries with different national social to healthcare spending ratios: Minneapolis, Minnesota, United States (US), and Nijmegen, Netherlands (NL).

### Design

Cross-sectional survey study. We adapted the Social Quality Model (SQM) to organize twenty diverse factors into four categories: Living Conditions (formal-societal: e.g., housing, education), Social Embeddedness (informal-societal: e.g., social support, trust), Societal Embeddedness (formal-individual: e.g., access to care, legal aid), and Self-Regulation (informal-individual: e.g., physical health, resilience). We developed a survey comprising validated instruments to assess each factor. We administered the survey in-person or by mail between March 2017 and August 2018.

**Data Availability Statement:** All relevant data are within the manuscript and its Supporting Information files.

**Funding:** This study was supported by the Robert Wood Johnson Foundation grant number #73156 (PI BB), "Reframing healthcare: A mixed methods study of patient experiences and outcomes integrating social and medical services for chronic conditions." The funders had no role in data collection and analysis, decision to publish, or preparation of the manuscript.

**Competing interests:** The authors have declared that no competing interests exist.

## Outcomes

We used Cantril's Self-Anchoring Scale to assess overall QOL. We used backwards step-wise regression to identify factors within each SQM category that were independently associated with QOL among US and NL participants ($p < 0.05$). We then identified factors independently associated with QOL across all categories ($p < 0.05$).

## Results

367 CHF patients from the US (32%) and NL (68%) participated. Among US participants, financial status, receiving legal aid or housing assistance, and resilience were associated with QOL, and together explained 49% of the variance in QOL; among NL participants, financial status, perceived physical health, independence in activities of daily living, and resilience were associated with QOL, and explained 53% of the variance in QOL.

## Conclusions

Four formal and informal factors explained approximately half of the variance in QOL among patients with CHF in the US and NL.

## Introduction

Social determinants of health over the life course influence the risk of cardiovascular disease incidence and its trajectory. [1] A growing body of literature is converging around the concept that even the highest quality healthcare, delivered without consideration for social care, is insufficient to promote cardiovascular health and quality of life (QOL). [1–4] To promote health and high QOL, a paradigm describing the importance of integrating health and social services is emerging. However, this macro-level paradigm must be accompanied by more granular exploration of this phenomenon at the individual-level to enact meaningful integration and change. The availability of such integrated services at the individual-level is especially important for people living with debilitating chronic illness, including congestive heart failure (CHF). [5] In addition to the availability of formal health and social services (e.g., hospitals, food assistance), informal health and social services (e.g., caregiving, social support, self-management) are also necessary to support cardiovascular health and QOL. [2–4, 6] Indeed, the absence of these informal health and social services has been linked to higher cardiovascular mortality and incident heart failure, poorer physical functioning, and lower QOL. [7–10]

The Social Quality Model (SQM) provides a comprehensive overview of these formal and informal health and social factors that influence quality of daily life (Fig 1). [11] The SQM describes the conditions that enhance people's well-being, capacity, and potential, and enables them to shape their own circumstances and contribute to society. [12] SQM factors are grouped into four conditions across the two constituting dimensions of the SQM, from formal to informal (x-axis) and from societal to individual (y-axis): Living Conditions (societal-formal), Social Embeddedness (societal-informal), Societal Embeddedness (individual-formal), and Self-Regulation (individual-informal). [13] Assessing factors across the SQM may give important clues to frontline service providers as to what is necessary to support the health and QOL of patients. However, while various factors related to the four SQM conditions are associated with health outcomes and/or QOL, the relative importance of each of these factors in supporting an individual's QOL remains unknown. [12, 14] Further, access to and valuation of

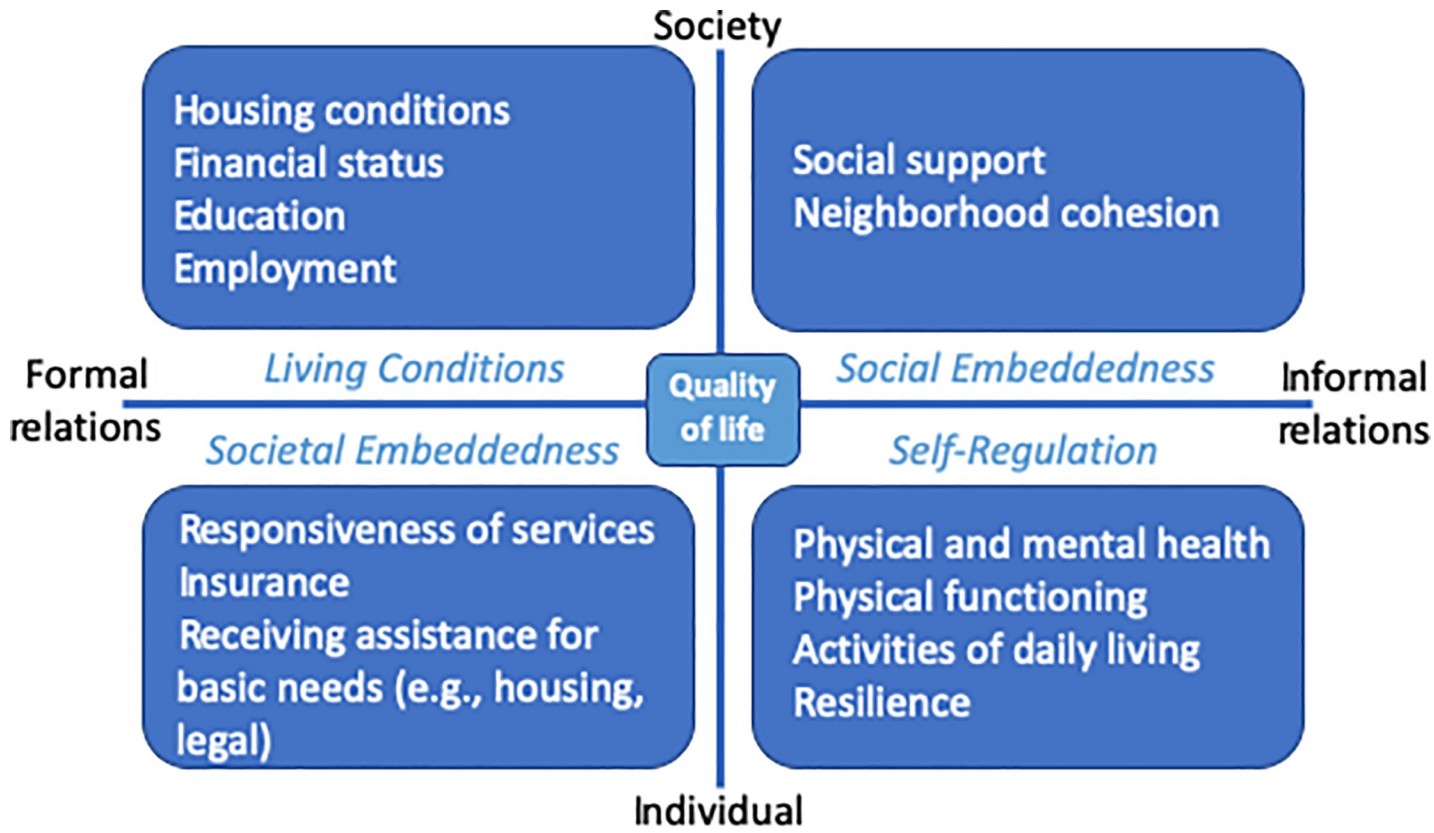

**Fig 1. The social quality model.** Theoretical framework describing four conditions necessary to support quality of life, adapted from Wolf (2016). [13]

these factors across the four SQM conditions may vary by sociocultural context and societal values and norms. For example, access to care may be more important to promote health and quality of life among those living in the United States (US) compared to countries where insurance coverage is nearly ubiquitous. Alternatively, responsiveness of services may be more important in countries where health insurance is universal and nationally organized. However, cross-national variation in individual residents' prioritization of the factors across the four SQM conditions has yet to be examined.

Societal valuation of health and social services may be inferred from the relative national spending on each. Bradley et al., (2011) reported that US spending from public sources on social services relative to healthcare services was significantly lower than all other countries in the Organization for Economic Cooperation and Development (OECD), suggesting that US public policy places lower relative value on social services compared to healthcare services. [15] Among OECD countries, higher social to healthcare service spending ratios were associated with better health outcomes, including longer life expectancy and lower infant mortality. It is plausible that these observed differences in national public spending are associated with differences in perceived access to and reported utilization of health and social resources, which in turn influence health and QOL at the individual level. If this logic holds, it is likely that these effects are magnified among individuals living with serious chronic illness due to greater proximate needs. Accordingly, we performed a cross-national survey study to characterize the presence or utilization of SQM factors, as well as to assess which combination of factors were independently associated with QOL among CHF patients living in two countries above and below the OECD average of social to health service spending ratios, the Netherlands (NL) and

the US, respectively. We hypothesized that differing cultural norms and values between the NL and US would affect individual-level valuation of SQM factors in supporting QOL.

## Methods

### Overview

We designed the Reframing Healthcare through the Lens of Coproduction (RHeLaunCh) study to explore healthcare and social care use and expenditure among patients with chronic disease living in the US or the NL. [16] We focused on CHF as a common, chronic condition that requires intensive healthcare and social care services for effective management, particularly in the later stages of disease. Our study combined quantitative and qualitative research methods, including (1) a literature scan; (2) a retrospective database study; (3) a survey study; and (4) a series of qualitative case studies. This paper describes our results from the cross-sectional survey study that assessed the prevalence of SQM factors and their association with QOL among a sample of patients living with CHF recruited from a single hospital site in the US and in the NL.

### Study setting

The study setting in the US was a cardiology clinic within a public, urban, university-affiliated, safety-net hospital, Hennepin Healthcare System (HHS) in Minneapolis, Minnesota. The study setting in the NL was an outpatient heart failure clinic within a public, academic health center, Radboud university medical center (Radboudumc) in Nijmegen, South Gelderland. Both sites are public institutions with similar missions (patient care, research, and education). At both sites, social services are not delivered directly via the healthcare system. Both sites have a fragmented system of institutions that deliver social services, though NL has more public funding to support these institutions compared with the US.

### Participant recruitment and survey administration

At both study sites, we used the following criteria for eligibility to participate in the study: 18 years of age or older, competent for consent, had a cardiologist-confirmed diagnosis of CHF, engaged in longitudinal care for CHF at a cardiology clinic at HHS or Radboudumc, and able to understand English in the US and Dutch or English in the NL. Patients with severe cognitive impairment or severe comorbid conditions (e.g., end-stage cancer) were excluded.

Participant recruitment and mode of survey administration varied between study sites due to a variety of factors, including population literacy levels. In the NL, a survey was mailed to all 447 eligible patients at their home address with an enclosed information letter that stated assistance for completing the questionnaire was available. After two weeks, non-responders received a reminder by mail. If, after four weeks, a potential participant did not respond to the mailed invitation, a nurse practitioner who works at the Radboudumc heart failure clinic reminded him or her to participate in the study at the time of a clinic visit. Patients who agreed to participate upon invitation from the nurse practitioner had the option of completing the questionnaire on paper or electronically, with or without assistance from a researcher.

In the US, study investigators screened patients who were scheduled for an in-person cardiology follow-up visit for study eligibility between March 2017 and August 2018. If eligible, the investigators called the patient prior to the visit to provide information about the survey study and invited him or her to participate. Any concerns about the patient's cognitive ability or other eligibility criteria were addressed in advance with the patient's provider (physician or advanced practice provider). The survey was administered by a trained researcher in the clinic

room in which the patient had just completed his or her cardiology consultation, or in another nearby office room. The questionnaire was also mailed to 266 eligible participants from HHS for sensitivity analyses to assess for differences in mode of administration.

Institutional review boards at HHS and Radboudumc approved this study. At Radboudumc, consent was implied if participants completed and mailed the survey. At HHS, verbal informed consent was obtained from all participants prior to survey administration. A scripted statement describing the purpose of the study, as well as how the data will be used, the confidential nature of the data, approximate time to complete the survey, the voluntary nature of the survey, and reassurance that participating or not participating would not influence their care at the clinic, was read to the participant by the in-person surveyor prior to beginning the survey. Contact information for the principal investigator was also given. If the patient did not want to participate, the survey was not completed. Completion of the survey implied that consent was given. The only persons present in the room at the time of verbal consent were the in-person surveyor and the subject.

## Patient and public involvement

Prior to the start of this study, we hosted conversations with people living with heart failure and their caregivers in the US and NL to explore factors they felt contributed to their QOL and their ability to manage their chronic disease. Based on these discussions, we adapted the social quality model to categorize the variety of formal and informal supports patients reported affected their QOL and disease management capability.

## Survey measures

The SQM provides a comprehensive overview of a variety of formal and informal individual and societal factors that influence quality of daily life across four conditions (Fig 1). We adapted existing, validated measures to assess factors related to each of the four SQM conditions. We used overall QOL as measured by current life evaluation as a proxy measure for quality of daily life.

First, American and Dutch experts in the fields of social determinants or social quality performed a literature scan and identified relevant measures or subscales that assess factors within each SQM condition. Then, all measures assessing a particular factor were discussed among a subgroup of study team members, considering face and content validity, length, internal consistency, availability of the measure in English and Dutch, and accessibility. We achieved consensus on inclusion of a single measure for each factor in the final survey (Table 1). We used the forward-backward translation method to translate measures only available in English to Dutch. [17] The resulting questionnaire was pilot tested in both study sites on a small sample of the target population to evaluate experiences administering the questionnaire as well as the understandability and feasibility of the questionnaire (i.e., clarity and relevance of items, length of questionnaire, order of items). Based on these findings, we made appropriate changes, resulting in two final questionnaires (i.e., NL (S1 File) and US (S2 File) versions). Both versions included the same measures and content, but the order of items differed to maintain optimal psychometric properties within each setting (e.g., in the US, demographic questions were placed towards the end to prevent influence on subsequent responses). [18]

Our primary outcome was overall QOL, measured using Cantril's Self-Anchoring Scale. [28] This measure assesses perceived overall QOL, rated using the visual of a ladder with rungs numbered from zero (worst possible QOL) to ten (best possible QOL). Cantril's Self-Anchoring Scale has been adopted by the Gallup World Poll and has been used in all OECD countries

**Table 1. Measures adapted to assess each factor within the social quality model.**

| SQM Quadrant | SQM Factor | Measure |
|---|---|---|
| Living Conditions | Housing conditions | Lehman Scale (living situation subscale) [19, 20] |
| | Perceived financial status | MacArthur Network Sociodemographic Questionnaire [21] |
| | Education | MacArthur Network Sociodemographic Questionnaire |
| | Employment | MacArthur Network Sociodemographic Questionnaire |
| Social Embeddedness | Social support | Duke Social Support Scale [22] |
| | Neighborhood cohesion | Collective Efficacy (social cohesion subscale) [23] |
| Societal Embeddedness | Responsiveness of services | CollaboRATE [24] |
| | Insurance status | RHeLaunCh team created |
| | Receiving assistance for basic needs | RHeLaunCh team created |
| Self-Regulation | Physical health | Lehman Scale (health subscale) [19, 20] |
| | Mental health | Behavioral Risk Factor Surveillance System |
| | Physical functioning | QoL Respiratory Illness Questionnaire (daily and domestic activities) [25] |
| | Activities of daily living | Activities of Daily Living [26] |
| | Resilience | Dutch Empowerment Scale [27] |

and found to have good psychometric properties across populations, with a Cronbach's alpha of 0.76 at the individual level and 0.81 at the national level. [29]

## Analysis

We first compared sociodemographic characteristics and health and social quality factors using t-tests or Chi-squared tests, depending on the scale of the characteristics and factors. Next, we assessed whether the number of reported social or healthcare services utilized were correlated with QOL using Pearson's correlation coefficients. Then, we used backwards step-wise regression to first identify factors that were significantly independently associated with QOL among US and NL participants within each SQM condition. We then combined the variables that were significantly and independently associated with QOL within each condition into a final model to identify variables that were significantly and independently associated with QOL across all conditions. In sensitivity analyses, we repeated these steps stratified by mode of administration (i.e., mailed versus administered in-person within each site's sample) and by level of education. Educational systems and standard levels of achievement differ across the US and NL. Therefore, we stratified by attainment of high school education because this was identified as a common threshold in both countries after discussions by Dutch and US researchers and reviewing the US and Dutch educational system.

Data were analyzed using Statistical Package for the Social Sciences (v22.0 for Windows, SPSS Inc., Chicago, IL, USA) and Stata SE (v14.1, College Station, TX). Statistical significance was set at $p < 0.05$.

## Results

Of the 447 questionnaires mailed to patients from Radboudumc, 249 were returned, resulting in a response rate of 55.7%. Of these, 23 questionnaires had missing data and were excluded from analyses, resulting in a sample size of 226 NL participants. Approximately 75% of patients invited by phone by HHS researchers to participate in the study on the day of their clinic visit agreed (N = 118). Of the 266 questionnaires mailed to patients from HHS, 20 were returned (response rate 7.5%).

**Table 2. US and NL participant demographic, clinical, and social characteristics.**

| Item | NL (N = 226) | US (N = 118) | P-value |
|---|---|---|---|
| *Demographic characteristics* | | | |
| Age (mean) | 66.1 | 62.9 | 0.05 |
| Gender (%Female) | 31.7 | 29.8 | 0.73 |
| Marital status (%married) | 64.4 | 26.3 | <0.001 |
| Home ownership (%) | 52.3 | 27.1 | <0.001 |
| Live alone (%) | 22.1 | 39.8 | <0.001 |
| Less than high school education (%) | 31.8 | 70.3 | <0.001 |
| Employed (full or part time) (%) | 16.1 | 22.9 | 0.11 |
| Retired (%) | 51.2 | 30.5 | <0.001 |
| *Clinical characteristics* | | | |
| Comorbidity* (%) | 59.8 | 98.9 | <0.001 |
| Polypharmacy** (%) | 75.5 | 98.9 | <0.001 |
| Any tobacco use (%) | 15.7 | 44.3 | <0.001 |
| Any alcohol use (%) | 40.0 | 22.6 | 0.001 |
| Any illicit drug use (%) | 1.7 | 22.6 | <0.001 |
| Physical health (%fair/poor) | 44.5 | 50.4 | 0.29 |
| Unable to walk up stairs (%) | 15.6 | 32.2 | <0.001 |
| Mental health (%fair/poor) | 10.2 | 22 | 0.002 |
| *Social characteristics* | | | |
| Social support (%"a lot") | 51 | 26.5 | <0.001 |
| My life has purpose (%Excellent/Very Good) | 92.0 | 92.8 | 0.81 |
| Trust neighbors (%Agree/Strongly Agree) | 91.4 | 71.7 | <0.001 |
| Resilience (mean) | 3.92 | 4.19 | <0.001 |

*Comorbidity was defined as being under treatment by a medical doctor for one or more health problems other than heart failure

**Polypharmacy was defined as use of ≥6 different types of prescribed medication

## Participant characteristics

Approximately one-third of the US and NL participants were women (Table 2). More NL participants were married, owned a home, had at least a high school education, and were retired. More US participants used tobacco, alcohol, and illicit drugs, and reported fair or poor mental health. There was no difference between groups in self-reported fair or poor health, but more US participants reported at least one other medical condition and had poorer functional status. More NL participants reported "a lot" of informal social support and were more trusting of neighbors. Both groups reported a strong sense of purpose in life.

## Overall QOL and correlated health and social service utilization

NL participants reported higher QOL than US participants (mean 7.12 vs. 6.5; p = 0.001). US participants reported using more social and healthcare services than NL participants. US participants reported using a mean (standard deviation (SD)) of 2.1 (1.9) social services and 3.9 (1.0) healthcare services, while NL participants reported using a mean of 1.1 (1.6) social services and 2.5 (1.4) healthcare services. Among the NL sample, use of social care services was negatively correlated with overall QOL (r = -0.19; p<0.01). No correlation between healthcare or social care service utilization and QOL was noted in the US sample. Among both samples, healthcare and social care service utilization were positively correlated (r = 0.2; p<0.05).

**Table 3. US condition-specific models: Factors within each condition that are independently associated with quality of life.**

| Living Conditions (R² = 0.14) | | Social Embeddedness (R² = 0.22) | |
|---|---|---|---|
| *Factor* | *ß (p value)* | *Factor* | *ß (p value)* |
| Perceived financial status | 0.23 (0.001) | Social support | 1.57 (<0.001) |
| Housing conditions | 0.78 (0.041) | Neighborhood cohesion | 1.38 (0.002) |
| **Societal Embeddedness (R² = 0.20)** | | **Self-Regulation (R² = 0.30)** | |
| *Factor* | *ß (p value)* | *Factor* | *ß (p value)* |
| Responsiveness of services | 0.38 (<0.001) | Physical functioning | 0.34 (0.033) |
| Legal aid | -1.58 (0.008) | Resilience | 1.82 (<0.001) |
| Housing aid | 1.24 (0.014) | | |

## Independent associations between SQM factors and QOL

Among the US sample, two or three factors from each SQM condition were independently associated with QOL (Table 3). Together, these two or three factors within each SQM condition explained anywhere from 14% (Living Conditions) to 30% (Self-Regulation) of the variance in QOL. In our sensitivity analyses using data from US participants who returned the survey by mail, results were similar.

Among the NL sample, two or three variables from each SQM condition were also independently associated with QOL (Table 4). Together, these two or three factors within each SQM condition explained anywhere from 6% (Societal Embeddedness) to 49% (Self-Regulation) of the variance in QOL.

In our final models that included all factors independently associated with QOL within each condition, we found four variables from three SQM conditions explained approximately half of the variance in QOL in the US and NL samples (Tables 5 and 6). In the US sample, better perceived financial status ($\beta = 0.17$, p = 0.004; Living Conditions), not receiving legal aid ($\beta = -1.14$, p = 0.012) and receiving housing aid ($\beta = 0.93$, p = 0.029; Societal Embeddedness), and greater resilience ($\beta = 2.43$, p<0.001; Self-Regulation) were independently associated with higher QOL; together, these variables explained 49% of the variance in QOL. In the NL sample, better financial status ($\beta = 0.18$, p<0.001; Living Conditions), greater resilience ($\beta = 0.96$, p<0.001), better perceived physical health ($\beta = 0.77$; p<0.001) and greater independence in activities of daily living ($\beta = 0.18$, p = 0.002; Self-Regulation) were independently associated with higher QOL; together, these variables explained 53% of the variance in QOL.

**Table 4. NL condition-specific models: Factors within each condition that are independently associated with quality of life.**

| Living Conditions (R² = 0.19) | | Social Embeddedness (R² = 0.16) | |
|---|---|---|---|
| *Factor* | *ß (p value)* | *Factor* | *ß (p value)* |
| Housing conditions | 0.51 (0.035) | Social support | 0.95 (<0.001) |
| Perceived financial status | 0.28 (<0.001) | Neighborhood cohesion | 0.58 (0.008) |
| Employment | 0.48 (0.043) | | |
| **Societal Embeddedness (R² = 0.06)** | | **Self-Regulation (R² = 0.49)** | |
| *Factor* | *ß (p value)* | *Factor* | *ß (p value)* |
| Responsiveness of services | 0.12 (0.003) | Physical health | p<0.001 |
| | | Fair/poor | -0.83 |
| | | Good | -0.33 |
| | | Very good/excellent | (ref) |
| Financial support | -0.54 (0.04) | Activities of daily living | 0.22 (<0.001) |
| | | Resilience | 1.00 (<0.001) |

**Table 5. US final model: SQM factors that are independently associated with quality of life ($R^2 = 0.49$).**

| Living Conditions | | Social Embeddedness | |
|---|---|---|---|
| *Factor* | *ß (p value)* | *Factor* | *ß (p value)* |
| Perceived financial status | 0.17 (0.004) | - | - |
| **Societal Embeddedness** | | **Self-Regulation** | |
| *Factor* | *ß (p value)* | *Factor* | *ß (p value)* |
| Legal aid | -1.14 (0.012) | Resilience | 2.43 (<0.001) |
| Housing aid | 0.93 (0.029) | | |

After pooling data from both countries and stratifying by attainment of high school education, results were largely similar, but neighborhood cohesion remained independently associated with QOL in our combined models including factors from all SQ conditions. Among those who did not complete high school (n = 47), better perceived financial status ($\beta = 0.33$, p = 0.004; Living Conditions), stronger neighborhood cohesion ($\beta = 1.15$, p = 0.024; Social Embeddedness), and greater resilience ($\beta = 1.25$, p = 0.016; Self-Regulation) were associated with higher QOL. Among those who had at least a high school education (n = 320), better perceived financial status ($\beta = 0.21$, p<0.001; Living Conditions), stronger neighborhood cohesion ($\beta = 0.53$, p = 0.002; Social Embeddedness), and better daily functioning ($\beta = 0.24$, p<0.001) and higher resilience ($\beta = 1.10$, p<0.001; Self-Regulation) were associated with higher QOL.

## Discussion

We explored associations between a wide range of social and healthcare factors with higher QOL among CHF patients recruited from two countries with markedly different societal spending patterns on social and healthcare services. In both the US and NL samples, four formal and informal factors explained approximately half of the variation in perceived QOL. Though we hypothesized that the different societal contexts would result in a different set of factors associated with QOL, we found that higher perceived financial status and resilience were independently associated with higher QOL among both samples. Our results do suggest, however, that between country differences also exist in the valuation of social service utilization and perceived health.

We inferred that societal public spending patterns reflected sociocultural norms and values regarding the accessibility of formal healthcare and social services, as well as informal health and social care resources. We hypothesized that these differing cultural norms and values may

**Table 6. NL final model: SQM factors that are independently associated with quality of life ($R^2 = 0.53$).**

| Living Conditions | | Social Embeddedness | |
|---|---|---|---|
| *Factor* | *ß (p value)* | *Factor* | *ß (p value)* |
| Perceived financial status | 0.18 (<0.001) | - | - |
| **Societal Embeddedness** | | **Self-Regulation** | |
| *Factor* | *ß (p value)* | *Factor* | *ß (p value)* |
| - | - | Physical health | p<0.001 |
| | | • Fair/poor | -0.77 |
| | | • Good | -0.31 |
| | | • Very good/excellent | (ref) |
| - | - | Activities of daily living | 0.18 (0.002) |
| | | Resilience | 0.96 (<0.001) |

affect individual-level valuation of these resources. However, the US participants had higher rates of utilization of both healthcare and social services and use of these resources was not associated with higher QOL. Among the NL sample, higher utilization of social services was negatively correlated with QOL. Rather, use of informal social resources, including higher social support and neighborhood cohesion, were both rated higher in the NL than the US and were associated with higher QOL. These findings suggest that the collective cultural ethos in the NL places greater value on the support from informal social resources to foster QOL. While this extends to societal provision of formal social services, if one needs to use these resources, it is associated with a poorer QOL in the NL perhaps because the higher valued informal resources are unavailable. Decentralization of formal social services in the NL in 2015 has been reported to have resulted in greater difficulty accessing services as well as poorer quality of services. [30]

Our findings support the need for integrated social and health programs to promote QOL among people living with serious chronic illness. [1, 5] There are several examples of programs that have successfully achieved better health outcomes, including fewer hospital readmissions, better physical functioning, fewer depressive symptoms, and lower costs of care, through the implementation of interdisciplinary teams that support social and health care needs. These programs also are more holistic and patient-centered, improving the QOL of the patients they serve. Our results are aligned with and augment these findings by also highlighting that informal social support systems and personal resources, such as resilience, should also be fostered to further improve QOL.

Perceived financial status and resilience were independently associated with QOL among participants from both countries. The link between higher income and QOL up to an annual gross income of approximately $70,000 has been well-established, and our results are consistent with the existing literature that higher perceived financial status is associated with better QOL. [31–33] Emerging literature also reports that greater resilience is linked to higher QOL among older adults and among individuals living with cancer. [34–37] It has been noted that the presence of positive social relationships and community integration at least partially mediates this relationship. [34] Related factors within the Social Embeddedness condition, measured by social support and neighborhood cohesion in our study, were significantly associated with QOL in the condition-specific analyses and in the pooled education-stratified analyses, but they were not independently associated with QOL in the final model once resilience and financial status were included. This suggests that in our sample, these SQ factors may also act as mediators of the relationship between resilience and QOL, but this relationship remains to be further explored.

In the NL, better perceived physical health and independence in activities of daily living were associated with higher QOL. These findings contradict those from a meta-analysis performed by Smith, et al., who reported that mental health was more closely related to QOL than physical health or physical functioning. [38] Similarly, a study performed in a large US sample reported weak correlations between QOL and perceived health. [31] It is possible that there are cultural, political, and environmental differences between the US and the NL that explain these differences. Around the time of this study, the Social Support Act (2015), a strong political movement in the NL to promote living as long as possible in the home, was gaining traction, suggesting high societal value of maintaining the physical capability to age in the community. In addition, the community environment in Nijmegen and its surrounding area is typically accessed by walking or biking, and thus, opportunities to participate and engage in meaningful activities require physical functioning. It has been reported that over half of the Dutch population participates in sports or engages in other physical activity at least once weekly, and that the Dutch highly value physical participation as well as having enough vitality

to maintain independence. As such, it is plausible that impediments to physical health have a more negative impact on QOL in the NL than in the US.

Our sample in the US had fewer material and immaterial resources than the population in the NL. This group was less educated and less likely to be married or partnered. These factors may have resulted in the US sample's lower reported quality of life as they affect an individual's ability to cope with daily life, enjoy a dignified life, and access opportunities. [39] As such, these sociodemographic characteristics may also have influenced the factors that we found to be independently associated with quality of life among this sample including factors in the social embeddedness quadrant, such as receiving legal and housing aid. Populations with lower education and income also tend to have fewer or weaker social relationships and tend to be embedded in neighborhoods with lower social cohesion, [40] and are therefore less protected by personal ties (e.g., being married is a protective factor for health and well-being). This has implications for quality of life because social support is an important source for well-being and vital in ensuring health (Cobb, 1976; Vaux, 1985). [41, 42] This may be the underlying reason that neighborhood cohesion was independently associated with quality of life among the subgroup that had less than a high school diploma or equivalent in our secondary analyses.

Nevertheless, the fact that the social quality factors that were independently associated with QOL in our final models originated from more than one SQ condition, and that together these factors explained half the variation in QOL, supports the theory underlying the Social Quality Model that formal and informal resources at the societal and individual levels are essential to promote QOL. This is aligned with other person-centered models that describe multiple domains of determinants of health and well-being. [43–45] No single resource is sufficient to maintain high quality of life, and individuals benefit from access to high quality resources in all four SQ conditions. As such, outcomes-driven policies that support multi-level, multi-sector change are more likely to foster high QOL among residents. [46, 47]

Our study had several limitations. First, people in the US sample were less educated. As such, our secondary analyses stratified by education shed some light on the similarities and differences in associations with QOL among more similar groups. Because of the differences in education level among our two populations, each site used a different primary mode of survey administration, which may have biased results. It is plausible that those who had the survey verbally administered would not want to disclose undesirable information, such as illicit drug use. However, in our sensitivity analysis among the US sample comparing verbally administered versus mailed responses, there was no difference in reported tobacco, alcohol, or illicit drug use by mode of administration, suggesting this had little impact. Second, participants in the US were more likely to report having at least one other medical condition and being prescribed at least six medications. As such, it is likely US participants were sicker than NL participants, but we did not have comparable data across sites on detailed clinical information, such as left ventricular ejection fraction, brain natriuretic peptide, device therapy, and specific comorbid medical conditions to further characterize these differences. Third, we were only able to test SQM factors for which we could find validated measures. It is possible that there are other unmeasured factors within the SQM that affect QOL, but the measures we included explained approximately half of the variance in QOL. Finally, our samples were recruited from a single site in the US and the NL, respectively. Though we chose these sites because they provided comparable access to health and social services, making our inferences about health and social service utilization more valid, these sites may not be representative of their respective country's typical health system. HHS is a public health system serving primarily low-income, Medicaid patients, yet it offers greater access to health and social service care compared to most US health systems. [48] This limits the generalizability of our results.

## Conclusions

This report adds to the literature by providing a comprehensive, patient-level assessment of the need for adequate integration of healthcare and non-healthcare, formal and informal services to support QOL among people living with severe chronic illness. Our findings may add further insight into the population health effects of observed international differences in spending patterns on healthcare and social services. In this cross-national survey study, including a site in the US and a site in the NL that provided similar access to health and social services, we found four factors from three conditions of the SQM were independently associated with QOL, and explained approximately half of the variation in QOL within each population. These factors were diverse, including societal and individual level factors, as well as formal and informal factors. Our findings provide individual-level data that support the call for integrated, holistic systems of care that address formal and informal health and social needs for people living with chronic illness.

## Supporting information

**S1 File. NL version of RHeLaunCH survey.**
(PDF)

**S2 File. US version of RHeLaunCH survey.**
(DOC)

**S3 File. Deidentified data file.**
(XLSX)

**S1 Fig. Original version of social quality model.**
(PDF)

## Acknowledgments

We would like to acknowledge the valuable contributions of Syl Jones and Joy McAvoy to this study as well as the larger RHeLaunCH project. Syl organized focus groups and discussions among CHF patients that informed the content of this study. He also administered surveys to many of the HHS participants included in this study. Joy provided organizational oversight for the entire RHeLaunCH project. We genuinely appreciate their dedication to this work and their respective contributions.

## Author Contributions

**Conceptualization:** Brita Roy, Judith R. L. M. Wolf, Michelle D. Carlson, Bradley Bart, Paul Batalden, Julie K. Johnson, Hub Wollersheim, Gijs Hesselink.

**Data curation:** Brita Roy, Michelle D. Carlson, Bradley Bart, Hub Wollersheim, Gijs Hesselink.

**Formal analysis:** Brita Roy, Reinier Akkermans, Gijs Hesselink.

**Funding acquisition:** Brita Roy, Bradley Bart, Paul Batalden, Julie K. Johnson, Hub Wollersheim.

**Investigation:** Julie K. Johnson, Hub Wollersheim.

**Methodology:** Brita Roy, Judith R. L. M. Wolf, Bradley Bart, Paul Batalden, Hub Wollersheim, Gijs Hesselink.

**Resources:** Bradley Bart.

**Supervision:** Bradley Bart, Paul Batalden, Hub Wollersheim.

**Writing – original draft:** Brita Roy, Gijs Hesselink.

**Writing – review & editing:** Brita Roy, Judith R. L. M. Wolf, Michelle D. Carlson, Reinier Akkermans, Bradley Bart, Paul Batalden, Julie K. Johnson, Hub Wollersheim, Gijs Hesselink.

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
