## [Decision Letter · Decision Letter 0]

6 Jan 2020

PONE-D-19-33707

An international comparison of factors affecting quality of life among patients with congestive heart failure: A cross-sectional study

PLOS ONE

Dear Dr. Roy,

Thank you for submitting your manuscript to PLOS ONE. After careful consideration, we feel that it has merit but does not fully meet PLOS ONE’s publication criteria as it currently stands. Therefore, we invite you to submit a revised version of the manuscript that addresses the points raised during the review process.

Both reviewers raised significant concerns about the validity of the results and their interpretation. Among others, additional data are required and extensive revision is needed before the manuscript may be considered. In particular, the cohorts need te be much better described. In addition, analyses must consider the differences in the populations.

If you are unable to add these additional results, the manuscript is not acceptable for publication. In this case, please let us know that you will not resubmit a revised manuscript. If you are convinced to be able to address all questions raised and to provide the additional data, we are happy to re-review your manuscript.

We would appreciate receiving your revised manuscript by Feb 20 2020 11:59PM. To enhance the reproducibility of your results, we recommend that if applicable you deposit your laboratory protocols in protocols.io, where a protocol can be assigned its own identifier (DOI) such that it can be cited independently in the future. For instructions see: http://journals.plos.org/plosone/s/submission-guidelines#loc-laboratory-protocols

We look forward to receiving your revised manuscript.

Kind regards,

Hans-Peter Brunner-La Rocca, M.D.

Academic Editor

PLOS ONE

Journal Requirements:

2. Please provide additional details regarding participant consent.

In the ethics statement in the Methods and online submission information, please ensure that you have specified (a) whether consent was informed and (b) what type you obtained (for instance, written or verbal, and if verbal, how it was documented and witnessed).

If your study included minors, state whether you obtained consent from parents or guardians.

If the need for consent was waived by the ethics committee, please include this information.

Reviewers' comments:

Reviewer's Responses to Questions

**Comments to the Author**

1. Is the manuscript technically sound, and do the data support the conclusions?

Reviewer #1: No

Reviewer #2: Partly

2. Has the statistical analysis been performed appropriately and rigorously? 

Reviewer #1: Yes

Reviewer #2: Yes

3. Have the authors made all data underlying the findings in their manuscript fully available?

Reviewer #1: No

Reviewer #2: No

4. Is the manuscript presented in an intelligible fashion and written in standard English?

Reviewer #1: Yes

Reviewer #2: Yes

5. Review Comments to the Author

Reviewer #1: Review of “An international comparison of factors affecting quality of life among patients with

congestive heart failure: A cross-sectional study” by Roy et al.

This study explored associations among twenty formal and informal, societal and individual-level factors and quality of life (QOL) among people living with HF in two settings (NL and US) with different healthcare and social care systems and sociocultural contexts. At the end, four formal and informal factors explained approximately half of the variance in QOL among patients with HF in the US and NL.

This study deepens into the concept that even the highest quality healthcare, delivered without consideration for social care, is insufficient to promote cardiovascular health and quality of life. This is a very well-designed and conducted study in an increasingly relevant topic. Methodologically clean at the pre-survey level.

The survey was conducted at 2 public institutions with similar missions (patient care, research, and education). At both sites, social services are delivered not via the healthcare system but via a fragmented system of social institutions. It would be worthwhile to know what would have been the results if a private US institution, which is the paradigm of US healthcare, was included in the study. I’m uncertain whether these sites are representative of their respective country’s typical health system.

Comments:

- 55.7% returned questionnaires in the NL, only 7.5% in the US. Most of the US participants completed the survey in their clinic visit, which is a different setting. This is a major bias of the data reported.

- People in the US were sicker and less educated, both variables with important influence in the reported data.

- Table2 – A relevant demographic item, such as age is missing.

- 22.6% of illicit drug use beyond tobacco and alcohol in the US seems quite extreme.

- Please provide data on aucasic, afro-american or Hispanic origin.

- Figure 1 is the same as found in reference 16.

Reviewer #2: In the present paper the authors looked at the association between formal and informal, societal and individual-level factors and quality of life (QOL) among patients with heart failure. The analysis was based on a battery of established questionnaires and assessment of QOL by the Cantril Ladder, an established QOL scale. To evaluate the role of different healthcare and social care systems and sociocultural contexts on the relative contribution of formal and informal, societal and individual-level factors on QOL two cohorts in the US (n=118) and the Netherlands (n=226) were studied. The authors report the following key findings: financial status and resilience were associated with QOL in both cohorts. Among US participants, receiving legal aid or housing assistance were also associated with QOL, and among NL participants, perceived physical health and independence in activities of daily living were in addition associated with QOL. In both cohorts approximately 50% of the variance of QOL were explained by the assessed between formal and informal, societal and individual-level factors.

General comment: This study looks at an interesting aspect, and the findings deserve attention. However, the interpretation of the results is not easy for several reasons. First, already in Table 2 it becomes evident that the cohorts differed significantly. Participants were not only more often married but also more often retired (age is not reported!). Thus, in the different periods of life different aspects may have different impact on QOL. Therefore, the impact of the place of living and the healthcare and social care system is hard to discriminate given these differences between groups. Furthermore, we know that patients had “heart failure” but no details are reported. To understand the results, we need information on LVEF, NYHA class, BNP/NT-proBNP, exercise capacity, medication and device therapy as well as comorbidities. There is some vague information on comorbidities and medication. However, “polypharmacy” is typically present in heart failure patients, and this is not necessarily bad. Insufficient or overtreatment as well as comorbidities including orthopedic or psychiatric diseases (alcohol or illicit drug dependence) may be relevant. I think that the study we benefit from additional information on these aspects.

6. PLOS authors have the option to publish the peer review history of their article (what does this mean?). If published, this will include your full peer review and any attached files.

Reviewer #1: No

Reviewer #2: No

---

## [Author Response · Author response to Decision Letter 0]

24 Feb 2020

Dear Dr. Brunner-La Rocca,

Thank you for your careful consideration of our manuscript titled, “An international comparison of factors affecting quality of life among patients with congestive heart failure: A cross-sectional study” (PONE-D-19-33707), for publication in PLOS ONE. My co-authors and I appreciate the reviewers’ comments and we welcome the opportunity to resubmit a revised version of the manuscript. We believe we have adequately addressed the reviewer’s concerns and the subsequent changes we have made to the manuscript have improved it. 

Below we have responded directly to each of the reviewers’ comments and we have specified where we have made changes to the text based on these comments. All line numbers refer to those in the file labeled ‘Revised Manuscript with Track Changes.’

Reviewer #1: 

1. This study explored associations among twenty formal and informal, societal and individual-level factors and quality of life (QOL) among people living with HF in two settings (NL and US) with different healthcare and social care systems and sociocultural contexts. At the end, four formal and informal factors explained approximately half of the variance in QOL among patients with HF in the US and NL. This study deepens into the concept that even the highest quality healthcare, delivered without consideration for social care, is insufficient to promote cardiovascular health and quality of life. This is a very well-designed and conducted study in an increasingly relevant topic. Methodologically clean at the pre-survey level. 

Response: This is an accurate summary of our study, and we are pleased the reviewer finds the topic interesting and relevant.

2. The survey was conducted at 2 public institutions with similar missions (patient care, research, and education). At both sites, social services are delivered not via the healthcare system but via a fragmented system of social institutions. It would be worthwhile to know what would have been the results if a private US institution, which is the paradigm of US healthcare, was included in the study. I’m uncertain whether these sites are representative of their respective country’s typical health system.

Response: It is true that only approximately one-quarter of US hospitals are public hospitals, like Hennepin Healthcare (HHS), and therefore may not be reflective of the majority, private healthcare system. We were limited to choosing only one site within each country, and therefore had to choose between internal versus external consistency. Because our goal was to assess the role of both social and health care, we chose two institutions with similar access to social services to enhance internal consistency. Likely, if a private US healthcare organization had been selected, access to social services would have been more limited than at HHS. We agree with the reviewer’s comment and we have acknowledged this limitation in the discussion section. That said, our primary findings – factors that affect quality of life among people living with severe chronic illness – are unlikely to be influenced by the type of healthcare institution a patient utilizes, but rather by access to other services that affect social quality.

3. 55.7% returned questionnaires in the NL, only 7.5% in the US. Most of the US participants completed the survey in their clinic visit, which is a different setting. This is a major bias of the data reported.

Response: The literacy level of the HHS population is much lower than that of the Radboudumc population. In addition, response rates to mailed surveys in the US have been declining steadily over the years. As such, the US study team opted to administer the survey during clinic visits to mitigate the threat of response bias among this sample. However, surveys were also mailed to a subset of the HHS population in order to assess whether differences based on mode of administration exist. Though the response rate to the mailed survey was very low among HHS patients, as anticipated, we found no consistent differences among responses to the mailed surveys compared to the administered surveys. Of note, the rate of reporting socially undesirable behaviors, such as drug and alcohol use, was no different among those who completed the survey by mail or administered by a healthcare professional. As such, we believe bias based on mode of administration is minimal, and we have noted this in the limitations section (lines 448-460).

4. People in the US were sicker and less educated, both variables with important influence in the reported data.

Response: We agree with this statement. We attempted to control as much as possible for these differences by incorporating variables that capture levels of education, health, and functioning in our models using backwards stepwise regression. This method identifies factors associated with quality of life, independent of all other factors in the model, including education, health, and physical functioning. Further, we performed stratified analyses among subgroups of higher and lower education, and explored independent associations of all social quality factors with quality of life within these groups to assess differences. Despite these efforts, we acknowledge there may be residual confounding, as we have stated in the limitations (lines 461-463). However, we have added new content to the discussion to further contextualize the interpretation of our findings among a group that was sicker and less educated (lines 423-436).

5. Table2 – A relevant demographic item, such as age is missing.

Response: Thank you for highlighting this unintended omission. We agree with the reviewer that age is an important influencer of heart failure. We have now included mean age in Table 2 (line 289).

6. 22.6% of illicit drug use beyond tobacco and alcohol in the US seems quite extreme.

Response: While one-fifth of the population using illicit drugs seems high, these rates are consistent with population estimates of the Minneapolis metropolitan statistical area. A report from SAMHSA in 2010a estimated 15.7% of the population over 12 years of age in Minneapolis-St. Paul used illicit drugs (the entire US average was 14.7%, in comparison). Because our sample was over 18 years of age and primarily low-income, and rates of opioid use have been steadily increasing across the US over the past decade, 22.6% is likely accurate.

a. https://www.samhsa.gov/data/sites/default/files/NSDUHMetroBriefReports/NSDUHMetroBriefReports/NSDUH-Metro-Minneapolis.pdf

7. Please provide data on aucasic, afro-american or Hispanic origin.

Response: While we can report these data for US participants, these race/ethnicity categories are not applicable in NL. Further, after World War II demonstrated the ease by which Nazis were able to identify persons of Jewish ancestry, most physicians in the NL refuse to code ethnicity. As such, ethnicity data are unreliable in the NL and we do not believe it would be valuable to include incomparable categories of race/ethnicity. 

8. Figure 1 is the same as found in reference 16.

Response: Figure 1 was adapted from the Wolf JR, et al. (2013) paper, which is reference 13. Our team initially published this adaptation of the figure in the paper describing the protocol for the larger grant of which this study is a part in reference 16, Hesselink, et al., (2017). Both are referenced in the manuscript, and if the editor wishes, we are open to modifying the order of references to clarify this.

Reviewer #2: 

1. General comment: This study looks at an interesting aspect, and the findings deserve attention. However, the interpretation of the results is not easy for several reasons. 

Response: We appreciate the reviewer’s interest in the study and in our findings. We have responded to the reviewer’s concerns about interpretation of results below.

2. First, already in Table 2 it becomes evident that the cohorts differed significantly. Participants were not only more often married but also more often retired (age is not reported!). Thus, in the different periods of life different aspects may have different impact on QOL. Therefore, the impact of the place of living and the healthcare and social care system is hard to discriminate given these differences between groups. 

Response: We agree that the sociodemographic characteristics of these samples were different. As noted above (Reviewer 1, Comment #5), we have now included age in Table 2. NL participants had a mean age of 66 years and US participants had a mean age of 63 years, though this difference was not statistically significant. As such, participants were generally in similar stages of life, though it is possible they had different resources available to them based on retirement status. Participants in the US did have fewer material and immaterial resources, however, and these can influence health and quality of life. Though we controlled for many of these differences using backwards stepwise regression (i.e., all factors in the final model were associated with quality of life independent of all other factors in the social quality model), residual confounding may still be present. We have added additional content to contextualize the differences in our cross-national sample to the discussion (Lines 423-436). 

3. Furthermore, we know that patients had “heart failure” but no details are reported. To understand the results, we need information on LVEF, NYHA class, BNP/NT-proBNP, exercise capacity, medication and device therapy as well as comorbidities. There is some vague information on comorbidities and medication. However, “polypharmacy” is typically present in heart failure patients, and this is not necessarily bad. Insufficient or overtreatment as well as comorbidities including orthopedic or psychiatric diseases (alcohol or illicit drug dependence) may be relevant. I think that the study we benefit from additional information on these aspects.

Response: Clinical diagnosis is the gold standard for diagnosis of heart failure, and all participants included in the study had a cardiologist-confirmed diagnosis of heart failure. This has now been clarified on lines 172-173. We have also included how we defined comorbidity and polypharmacy to clarify these descriptive statistics (Lines 290-292). We also have included alcohol and illicit drug dependence. Although we did not assess the NYHA class, we asked questions regarding physical functioning and exercise capacity that provide similar data (see Table 2). If the editor wishes, we can perform a chart review to obtain additional clinical data on specific medication and device therapy to include in Table 2 for HHS patients (this data is not available for Dutch participants), however, this would be a major departure from our theoretical framework. Because this study was not a clinical trial, and instead aimed to assess the impact of social and health care services on our primary outcome of quality of life, we do not believe these clinical data would impact the strength and relevance of our conclusions. Rather, functional and psychosocial factors assessed in the survey are likely to have a greater impact. 

Again, we thank you for the opportunity to revise our manuscript and we hope you will find this version suitable for publication in PLOS ONE. If you have any further recommendations on how to improve the manuscript, we welcome these suggestions and we will work hard to implement them. We believe this publication will be a valuable contribution to the literature, and we appreciate your time and consideration.

---

## [Decision Letter · Decision Letter 1]

17 Mar 2020

PONE-D-19-33707R1

An international comparison of factors affecting quality of life among patients with congestive heart failure: A cross-sectional study

PLOS ONE

Dear Dr. Roy,

Thank you for submitting your manuscript to PLOS ONE. After careful consideration, we feel that it has merit but does not fully meet PLOS ONE’s publication criteria as it currently stands. Therefore, we invite you to submit a revised version of the manuscript that addresses the points raised during the review process.

**I would like to ask the authors to include the lack of data mentioned by reviewer #2 as a limitation to the study.**

We would appreciate receiving your revised manuscript by May 01 2020 11:59PM. To enhance the reproducibility of your results, we recommend that if applicable you deposit your laboratory protocols in protocols.io, where a protocol can be assigned its own identifier (DOI) such that it can be cited independently in the future. For instructions see: http://journals.plos.org/plosone/s/submission-guidelines#loc-laboratory-protocols

We look forward to receiving your revised manuscript.

Kind regards,

Hans-Peter Brunner-La Rocca, M.D.

Academic Editor

PLOS ONE

Reviewers' comments:

Reviewer's Responses to Questions

**Comments to the Author**

1. If the authors have adequately addressed your comments raised in a previous round of review and you feel that this manuscript is now acceptable for publication, you may indicate that here to bypass the “Comments to the Author” section, enter your conflict of interest statement in the “Confidential to Editor” section, and submit your "Accept" recommendation.

Reviewer #1: All comments have been addressed

Reviewer #2: All comments have been addressed

2. Is the manuscript technically sound, and do the data support the conclusions?

Reviewer #1: Yes

Reviewer #2: Yes

3. Has the statistical analysis been performed appropriately and rigorously? 

Reviewer #1: Yes

Reviewer #2: Yes

4. Have the authors made all data underlying the findings in their manuscript fully available?

Reviewer #1: Yes

Reviewer #2: Yes

5. Is the manuscript presented in an intelligible fashion and written in standard English?

Reviewer #1: Yes

Reviewer #2: Yes

6. Review Comments to the Author

Reviewer #1: (No Response)

Reviewer #2: The authors have appropriately responded to the reviewers' comments. I was thinking that a better characterization of the patients in terms of measures of cardiac function, functional capacity, comorbidities and medication would have strengthened the paper. Obviously, this data is not available, and the authors argue that the findings can be properly interpreted even in absence of this information. I think that it is up to the editors to decide whether the paper is acceptable despite this Limitation.

7. PLOS authors have the option to publish the peer review history of their article (what does this mean?). If published, this will include your full peer review and any attached files.

Reviewer #1: No

Reviewer #2: No

---

## [Author Response · Author response to Decision Letter 1]

19 Mar 2020

Dear Dr. Brunner-La Rocca,

Thank you again for your consideration of our manuscript titled, “An international comparison of factors affecting quality of life among patients with congestive heart failure: A cross-sectional study” (PONE-D-19-33707), for publication in PLOS ONE. Below we have responded directly to the remaining editor’s and reviewer’s comments and we have specified where we have made changes to the text based on these comments. All line numbers refer to those in the file labeled ‘Revised Manuscript with Track Changes.’

Editor: I would like to ask the authors to include the lack of data mentioned by reviewer #2 as a limitation to the study.

Reviewer 2: The authors have appropriately responded to the reviewers' comments. I was thinking that a better characterization of the patients in terms of measures of cardiac function, functional capacity, comorbidities and medication would have strengthened the paper. Obviously, this data is not available, and the authors argue that the findings can be properly interpreted even in absence of this information. I think that it is up to the editors to decide whether the paper is acceptable despite this Limitation.

Response: We have added the following text to the limitations paragraph to directly address the constraints of our comparable data (lines 420-424): “participants in the US were more likely to report having at least one other medical condition and being prescribed at least six medications. As such, it is likely US participants were sicker than NL participants, but we did not have comparable data on detailed clinical information, such as left ventricular ejection fraction, brain natriuretic peptide, device therapy, and specific comorbid medical conditions to further characterize these differences.”

Again, we thank you for the opportunity to revise our manuscript and we hope you will find this version suitable for publication in PLOS ONE. 

Sincerely,

Brita Roy, MD, MPH, MHS

Assistant Professor of Medicine and Epidemiology

---

## [Editor Report · Decision Letter 2]

23 Mar 2020

An international comparison of factors affecting quality of life among patients with congestive heart failure: A cross-sectional study

PONE-D-19-33707R2

Dear Dr. Roy,

We are pleased to inform you that your manuscript has been judged scientifically suitable for publication and will be formally accepted for publication once it complies with all outstanding technical requirements.

With kind regards,

Hans-Peter Brunner-La Rocca, M.D.

Academic Editor

PLOS ONE
---

## [Editor Report · Acceptance letter]

26 Mar 2020

PONE-D-19-33707R2 

An international comparison of factors affecting quality of life among patients with congestive heart failure: A cross-sectional study 

Dear Dr. Roy:

I am pleased to inform you that your manuscript has been deemed suitable for publication in PLOS ONE. Congratulations! Your manuscript is now with our production department. 

With kind regards,

on behalf of

Dr. Hans-Peter Brunner-La Rocca 

Academic Editor

PLOS ONE